# Fishing for newly synthesized proteins with phosphonate-handles

Fleur Kleinpenning[1,2], Barbara Steigenberger[1,2], Wei Wu [1,2✉] & Albert J. R. Heck [1,2✉]

Bioorthogonal chemistry introduces affinity-labels into biomolecules with minimal disruption to the original system and is widely applicable in a range of contexts. In proteomics, immobilized metal affinity chromatography (IMAC) enables enrichment of phosphopeptides with extreme sensitivity and selectivity. Here, we adapt and combine these superb assets in a new enrichment strategy using phosphonate-handles, which we term PhosID. In this approach, click-able phosphonate-handles are introduced into proteins via 1,3-dipolar Huisgen-cycloaddition to azido-homo-alanine (AHA) and IMAC is then used to enrich exclusively for phosphonate-labeled peptides. In interferon-gamma (IFNγ) stimulated cells, PhosID enabled the identification of a large number of IFN responsive newly synthesized proteins (NSPs) whereby we monitored the differential synthesis of these proteins over time. Collectively, these data validate the excellent performance of PhosID with efficient analysis and quantification of hundreds of NSPs by single LC-MS/MS runs. We envision PhosID as an attractive and alternative tool for studying stimuli-sensitive proteome subsets.

[1] Biomolecular Mass Spectrometry and Proteomics, Bijvoet Center for Biomolecular Research and Utrecht Institute for Pharmaceutical Sciences, Utrecht University, Padualaan 8, 3584 CH Utrecht, The Netherlands. [2] Netherlands Proteomics Centre, Padualaan 8, 3584 CH Utrecht, The Netherlands. ✉email: w.wu1@uu.nl; a.j.r.heck@uu.nl

The proteome varies from cell to cell, but also very much over time and across different cellular states. While the total cellular proteome can nowadays be measured by mass spectrometry (MS) with quite some success in depth and precision[1,2], the consensus is that we still need specific capture methods when focusing on a selected smaller set of proteins of interest (POIs). These POIs could be a chemically-modified subset, or a newly-synthesized or a secreted subset of the total proteome, which may be very low abundant against the background of the full proteome. Novel methods in isolating and accurately quantifying these biologically important POIs over time will substantially improve our understanding of how cells respond to stimuli and maintain homeostasis.

To date, a plethora of methods to label and purify POIs have been developed by combining affinity handles with click chemistry. Click chemistry reactions allow pairs of functional groups to react rapidly and selectively ('click') with each other under mild, aqueous conditions, thereby allowing affinity handles to be added chemically to POIs for specific isolation[3–6]. Chemical 'clicks' can be based on protein-protein interactions (chemical/photo-crosslinking[7,8]; proximity-dependent labeling[9–11]), functional activity (activity-based protein profiling[12,13]), or can rely on protein synthesis (bioorthogonal noncanonical amino acid tagging, BONCAT[14,15]). The beauty of bioorthogonal chemistry lies in the minimal disruption to the original biological systems. For instance, BONCAT has been used to co-translationally introduce 'click-able' azides into proteins, to enable chemical ligation to alkyne-biotin and subsequent isolation via the biotin-streptavidin interaction[16]. These technical advances in purification of POIs were only possible with the advent of innovative tools in bioorthogonal tagging (e.g. methionine homolog azidohomoalanine, AHA)[14] and affinity enrichment strategies (e.g. streptavidin) devised to specifically purify these co-translationally tagged POIs[16]. The added flexibility to reverse affinity handle-probe pairs[14] and compatibility with pulsed-SILAC[17,18] have drastically expanded the utility of BONCAT to secretomics[19], degradomics[20], and many other innovative applications[21–24].

In many such click applications, biotin has been the affinity-handle for POI-retrieval, mainly due to its very strong non-covalent interaction with streptavidin[25]. While this minimizes loss of POIs even during stringent washing steps, elution from streptavidin can be problematic, as very harsh conditions (boiling, extreme pH or denaturants) are needed even for partial recovery of POIs. Such extreme conditions can also chemically modify and degrade residues in an unpredicted manner, which may preclude MS identification and hamper with the site-localization of post-translational modifications (PTM). Towards improving the recovery of captured biotinylated analytes, DiDBit (direct detection of biotin-containing tags) was developed to elute biotinylated peptides using NeutrAvidin, although this method was mostly used to enrich biotinylated proteins in vitro[26]. Other approaches include cleavable biotin probes as well as anti-biotin antibodies for direct detection of biotinylation sites[27]. Even then, the sensitivity of biotin-streptavidin interactions in cells is still limited by interference of endogenous biotinylated proteins (including several abundant cytoplasmic carboxylases)[28,29]. Collectively, these issues illustrate that there still is room for improvement and a need for better and a wider selection of affinity-handles for POI retrieval, ideally in combination with click chemistry.

Here, we introduce a novel enrichment strategy to purify proteins using 'click-able' phosphonate-handles, which can then be combined with a highly efficient and sensitive enrichment by immobilized metal affinity chromatography (IMAC)[30,31]; a method we term PhosID. We demonstrate that phosphonate-handles have superior enrichment specificity and sensitivity compared to functionalized biotin in a specific application to profile newly synthesized proteins. In addition, these phosphonate-handles can be synthesized directly by mixing commercially available chemicals in defined proportions. Endogenous phosphate groups on proteins can be removed by a broad-spectrum phosphatase, while retaining the phosphonate-label as they contain a stable P–C bond that is not cleaved by this enzyme[32]. Consequently, PhosID allows a near 100% clean and exclusive elution of peptides containing the bioorthogonal modification, thereby facilitating direct label-free quantification (LFQ) by LC-MS/MS. As such, we see PhosID as a significant addition to the repertoire of enrichment handles, for probing stimuli-sensitive and time-dependent proteome subsets.

## Results

**Preparing phosphonate-containing probes using bioorthogonal chemistry.** Bioorthogonal chemistry is widely applied to modify biomolecules without interfering with other functionalities present in living systems[3–6]. Typically, a chemical reporter incorporated in biomolecules reacts with a probe conjugated to the complementary functional group of this reporter. Two commonly used bioorthogonal reactions are the copper(I)-catalyzed azide-alkyne cycloaddition (CuAAC)[33,34] and the strain-promoted azide-alkyne reaction (SPAAC)[35]. CuAAC involves the Cu(I)-catalyzed reaction between an azide- and alkyne-functionalized molecule, whereas the SPAAC reaction is performed using an azide- and cyclooctyne-conjugated molecule without the need of a catalyst[36]. Both CuAAC and SPAAC result in the formation of a stable triazole linkage, which can then withstand harsh washing conditions during biomolecule isolation (Fig. 1a).

To further expand the selection of biorthogonal probes available to study new protein synthesis, we designed and synthesized three different phosphonate 'click-able' probes, each containing a phosphonic acid moiety attached to an alkyne (**P-alkyne**), azide (**P-azide**), or dicyclobenzooctyne (**P-DBCO**), for CuAAC and SPAAC reactions respectively (Fig. 1b). We envision that analytes modified by such phosphonate-containing probes may be retrieved efficiently even from complex lysate background by employing IMAC methods broadly in use for phospho-peptide enrichment (Fig. 1c).

**Enrichment of phosphonate-labeled peptides.** We prepared in situ the **P-alkyne**, **P-azide**, and **P-DBCO** probes from commercially available chemicals (Supplementary Fig. 1a–c) and tested the specificity of these probes on alkyne- or azide-functionalized bovine serum albumin (BSA). After trypsin digestion, BSA peptides clicked with respectively equimolar **P-alkyne**, **P-azide** or **P-DBCO** handles were retrieved by using an IMAC affinity setup that was previously optimized for phosphopeptides[37]. As shown in Fig. 2a, digested functionalized BSA 'clicked' efficiently with **P-alkyne**, **P-azide**, or **P-DBCO** probes, and the labeled peptides could be retrieved by IMAC enrichment with near-complete specificity (P-labeled peptides in green). About one-third of all BSA peptides were functionalized with the azide- or alkyne-handle (see input analysis in Supplementary Fig. 2), whereof 80–96% of the azide-modified and about 69% of the alkyne-modified peptides were labeled by the phosphonate-handle (green:blue ratio in input; Fig. 2a). In addition, less than 2.5% of the total peptide intensity in the input fraction of the peptides were found to be labeled with a partially hydrolyzed probe synthesis byproduct (Supplementary Fig. 2).

To further test the sensitivity of retrieving phosphonate-labeled peptides by IMAC, we functionalized BSA with decreasing molar-ratio of azide moieties to simulate a lower abundance of 'click-able' substrates (Supplementary Table 1). From this, we found

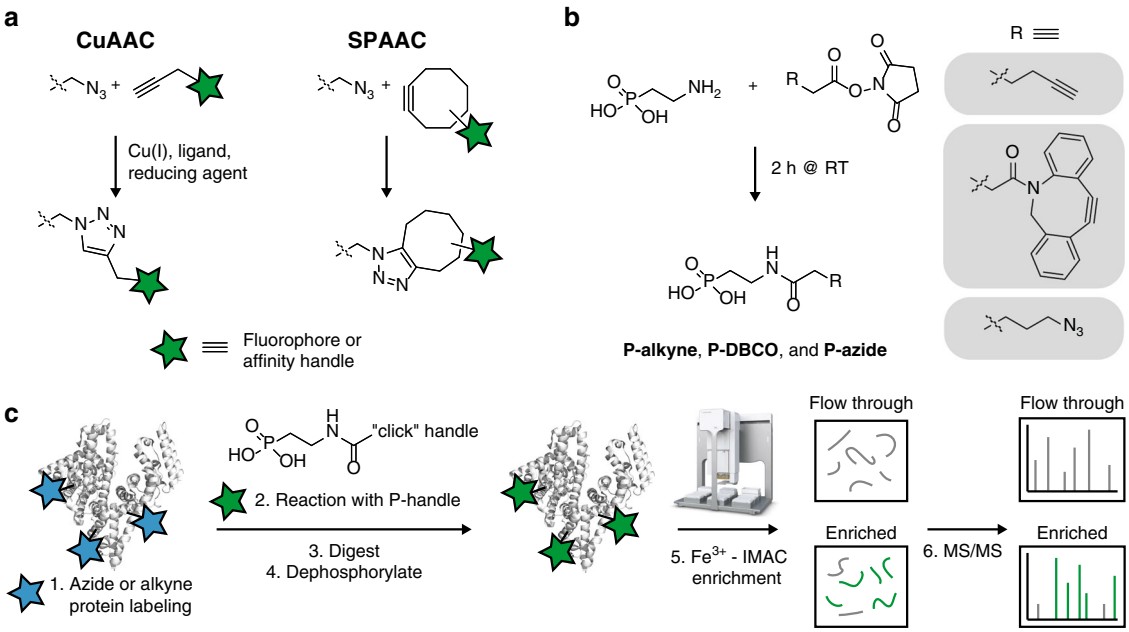

**Fig. 1 Phosphonic acid labeling of biomolecules and enrichment strategy. a** Scheme of CuAAC and SPAAC bioorthogonal reactions used in this study. **b** Synthetic route of the three different phosphonate-labeled probes. **c** General workflow for the phosphonate enrichment strategy (PhosID). Detailed experimental procedures and conditions are described in the "Methods" section.

that the phosphonate-labeling occurred largely proportional to the prevalence of azide handles, without compromising the enrichment specificity even at low substrate stoichiometry (Supplementary Fig. 3). To demonstrate that retrieval of P-labeled peptides from complex mixtures is still highly efficient, we spiked peptides of azide-functionalized BSA in decreasing ratios into an unlabeled HeLa digest (Supplementary Table 2). Following phosphatase treatment and IMAC enrichment, phosphonate-labeled BSA peptides were still well-detectable at a 1:10,000 (w:w) dilution ratio in the full HeLa digest (Fig. 2b). In addition, our data showed that the endogenous phosphate groups were effectively removed as the amount of identified phosphorylated peptides decreased from 10% to less than 1% upon increasing amounts of BSA (Supplementary Table 2). In other words, dephosphorylation of endogenous phosphopeptides and the recovery of phosphonate-labeled peptides by IMAC worked very well regardless of the ratio of phosphonate-labeled peptides to non-modified peptides.

Collectively these data illustrate the extreme sensitivity and specificity of the PhosID workflow and exemplify its promising application to retrieve low-abundant 'click-able' analytes from complex backgrounds. The PhosID strategy may thus provide an eminent alternative to the conventional biotin-streptavidin approach.

**Comparison of PhosID with biotin enrichment.** We next sought to benchmark the performance of PhosID against the most classical and widely used bead-based biotin-streptavidin pulldown approach (Fig. 3a). PhosID targets and allows direct identification and quantification of AHA-containing peptides, whereas in the biotin-streptavidin approach, biotinylated peptides typically are not detected in LC-MS/MS analysis due to the need to perform on-beads digestion. Therefore, it is difficult to make completely paired comparisons, but we still attempted to benchmark PhosID against the more conventional biotin-streptavidin approach, as the latter is currently still the most popular approach for affinity-clicked analytes. For this comparison, we prepared three kinds of

lysates as starting samples: (a) an unlabeled HeLa lysate, (b) a pulse-AHA labeled HeLa lysate (labeling for 24 h), and (c) a stable-AHA labeled HeLa lysate (10 splittings over 3 weeks), and used each time half of each lysate for enrichment by either PhosID or biotin. The goal was to establish the most direct comparisons of reproducibility, efficiency, specificity, and potential bias, by performing both strategies in parallel.

We first used an unlabeled HeLa lysate that did not contain click-able AHA residues to investigate the non-specific background binding. The data revealed that the PhosID approach exhibited a much lower non-specific background than the biotin approach (Fig. 3b, Supplementary Fig. 4). This low background in PhosID was even further reduced when the input material contained click-able AHA residues, with 96% of the proteins (2366 out of 2476) being identified by at least one peptide containing the expected Met → AHA substitution, as unambiguous evidence of AHA incorporation (Fig. 3c). On the other hand, the biotin approach was found to retrieve already >2000 non-specific background proteins from the unlabeled HeLa lysate. Streptavidin was not a major interfering contaminant in the biotin-streptavidin eluates, as peptides from streptavidin only contributed 0.5–3.5% to the total peptide intensity (Supplementary Data 2). Endogenous proteins with biotin cofactors on the other hand may contribute to this background, at least in part, as these will also bind and be retrieved by the streptavidin affinity purification. In contrast to the high specificity of PhosID, only about 1% of the proteins identified by using the streptavidin pulldown (17 out of 2601) had peptide-level evidence of Met → AHA incorporation (Supplementary Table 3). This observation is consistent with previous reports that quantification of biotin-enriched newly synthesized proteins (NSPs) still requires SILAC in combination for added confidence on NSP status[38,39]. Collectively, these data demonstrate that the PhosID approach performs favorably in enrichment specificity and is directly compatible with quantification at the level of AHA-modified peptides.

Since the PhosID approach enriches exclusively for AHA-containing peptides after digestion, we expected the sequence coverage of NSPs to be limited by the number of tryptic peptides containing methionine residues, unlike in the case of biotin

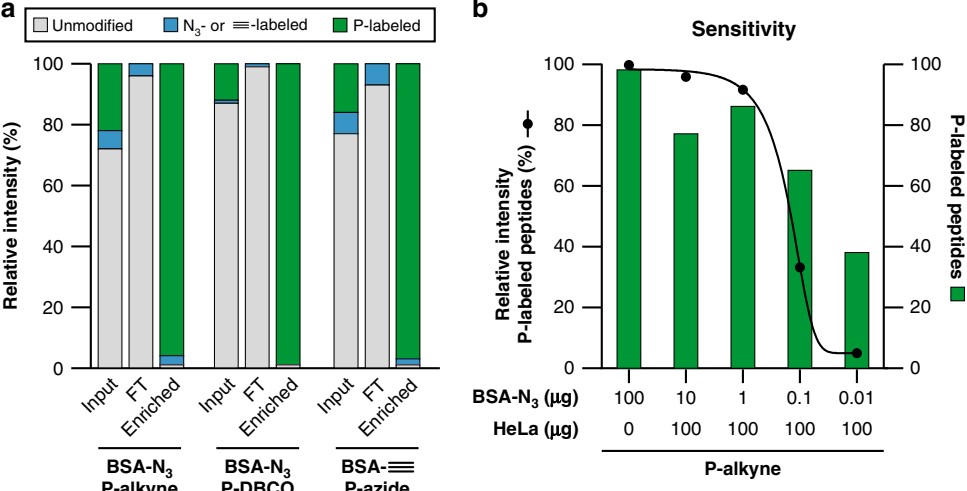

**Fig. 2 Labeling and enrichment of functionalized BSA by PhosID. a** Efficient PhosID enrichment as quantified by relative MS intensities. BSA was functionalized at the free amine groups of lysines to yield azide-modified BSA (BSA-N$_3$) or alkyne-modified BSA (BSA-N$_3$) and was clicked to the corresponding **P-alkyne/P-DBCO** or **P-azide**, respectively. After trypsin digestion, phosphonate-modified peptides were retrieved by IMAC purification. Relative intensity of BSA peptides detected in IMAC input, flowthrough (FT) and elution (enriched) fractions are plotted. Data from the control experiments are shown in Supplementary Fig. 2. **b** Sensitive retrieval of phosphonate-modified BSA peptides from complex Hela lysate. Digested BSA-N$_3$ was spiked in 100 μg of Hela digest in various proportions, clicked with **P-alkyne** and enriched by IMAC in the PhosID workflow as shown in Fig. 1c. The summed intensity of phosphonate-modified peptides are plotted relative to total intensity of all peptides detected in IMAC eluate (left axis). Green bars represent the number of unique phosphonate-labeled BSA peptides retrieved (right axis). Even at a BSA-N$_3$: HeLa ratio of 0.01:100 (ratio of 1:10,000; w/w), phosphonate-labeled BSA peptides were still well detectable. Source data provided in Source data file.

enrichment where digestion takes place after the enrichment at the protein level. This turned out not to be a concern, as a large number of PhosID proteins were still identified by 2–3 unique peptides (Fig. 3c), and 92% of all PhosID peptides could only be assigned to one protein (Supplementary Fig. 5). This is almost identical as found for the NSPs enriched by the biotin-streptavidin pulldown and digested thereafter (Supplementary Fig. 5). Given that the mean or median number of unique peptides per NSPs in PhosID does not improve further, even with long-term stable AHA-labeling (Supplementary Fig. 6), we conclude that this limit is largely imposed by the number of methionine residues that can be AHA-substituted in most proteins[40], and cannot be boosted further by any additional approach still involving methionine tagging.

In our experimental data, both the PhosID and the biotin-based enrichments were highly reproducible as shown by the strong correlation between the intensity of the peptides identified in the pulse-AHA replicates (Supplementary Fig. 7a, b). This correlation was even stronger when comparing the intensity at the protein level for phosphonic acid enriched pulse-AHA replicates (Supplementary Fig. 7c). In addition, PhosID does not bias for longer AHA labeling times, as pulse-AHA and stable-AHA labeled samples were also found to be highly correlated ($R^2$ coefficient 0.99 based on Pearson linear regression) (Fig. 3d). Since the quantification in the PhosID approach differs from that following biotin enrichment (using peptides with the label incorporated versus unmodified peptides in biotin-streptavidin approach), we compared instead the identity of the NSPs as determined by both methods. The overlap between proteins identified was moderate at 55% (Fig. 3e), although following PhosID we had unambiguous evidence of an incorporated and clicked AHA moiety at the peptide-level for all proteins detected, which was clearly not achievable with the biotin approach.

**Interferon gamma (IFNγ) responsive protein synthesis monitored by PhosID.** To demonstrate the utility of PhosID in a more relevant cellular context, we applied our optimized protocol to investigate Interferon-gamma (IFNγ) stimulation in two different cell lines. We adopted a classical AHA labeling workflow[41] to introduce 'click-able' azide functionalities into NSPs in HeLa and Jurkat cells, and then applied the PhosID strategy to identify NSPs induced by IFNγ in 4 or 24 h (Fig. 4a). To exclude housekeeping proteins that are constantly made regardless of stimuli, a paired control experiment without IFNγ stimulation was performed at every time point to allow background subtraction. Before performing the click reaction with the **P-alkyne** probe, we first successfully verified that IRF1 was indeed induced, by monitoring IRF1 abundance by western blot (Fig. 4b). IRF1 is a classical IFNγ-responsive gene downstream of STAT1, the main transcription factor orchestrating IFNγ pro-inflammatory responses[42,43].

After filtering for significant changes in protein abundance upon IFNγ stimulation against the unstimulated control, we observed the induction or suppression of 176 proteins in HeLa cells within 24 h (>2-fold, $p < 0.05$) (Fig. 4c). These significantly regulated NSPs included many protein products of well-documented interferon responsive genes (ISGs)[44,45], most notably the STATs, IFITs, and GBPs. Nearly all NSPs modulated by more than 10-fold in HeLa cells, 90% (19 out of 21 proteins), have been reported as interferon targets, in studies with extensive downstream verification and biological validations (see the annotated data points in Fig. 4c). This exemplifies the high selectivity and efficiency of our PhosID approach, and also confirms that PhosID data is reliable and recapitulative of knowledge acquired with conventional molecular biology methods.

In addition to the 21 proteins strongly influenced by IFNγ treatment, we also identified 155 NSPs that were significantly regulated by between 2 and 10-fold. Network analysis of these proteins assembled three dominant functional clusters in inflammation signaling, RNA splicing, and redox and metabolism[46,47], all of which are reportedly under the influence of IFNγ (Supplementary Fig. 8). A large majority of these targets are ISGs,

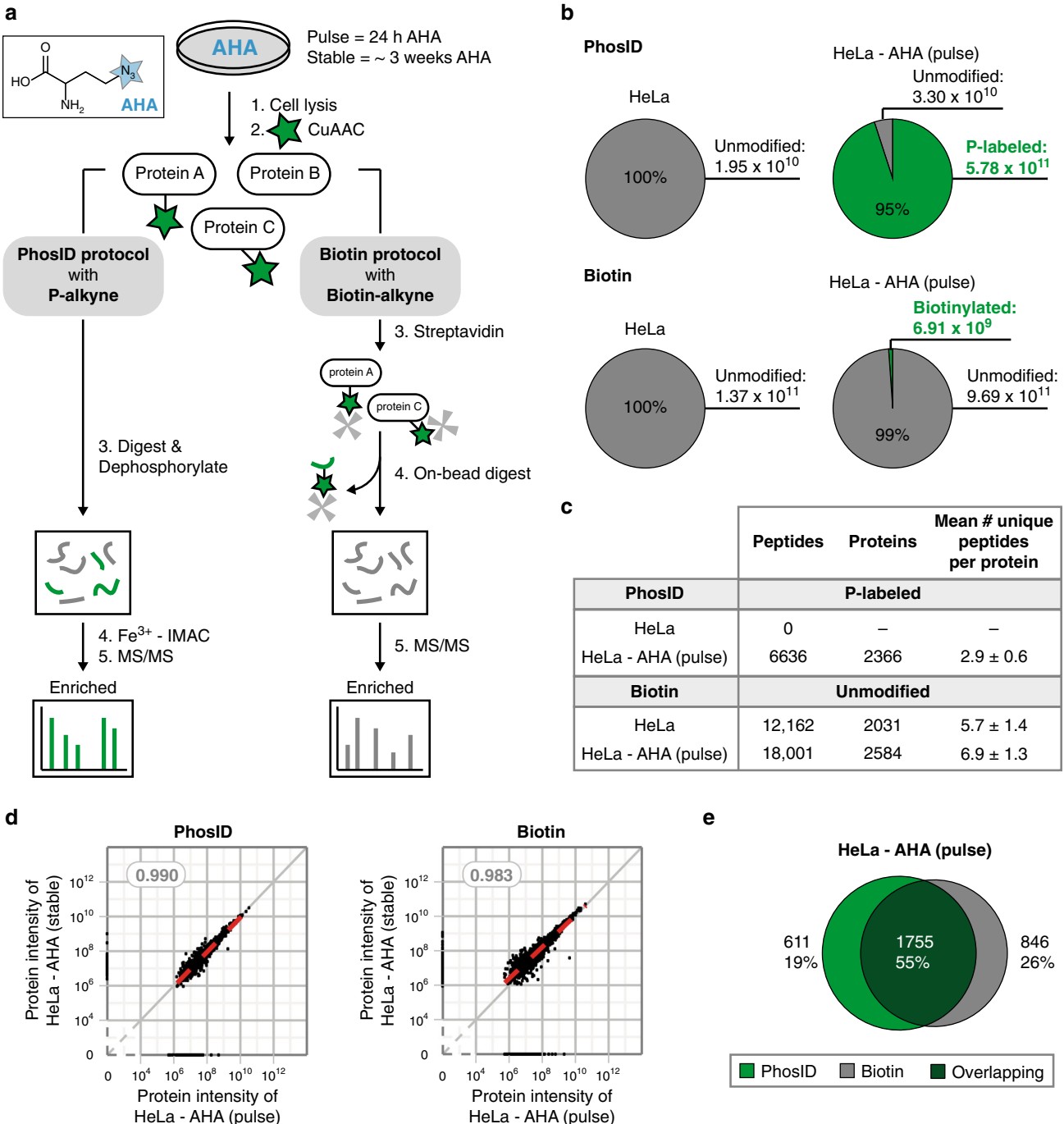

**Fig. 3 Enrichment of AHA-labeled proteins from HeLa cell by PhosID or biotin-streptavidin workflow. a** Parallel steps in PhosID and biotin-streptavidin workflows. To make comparisons between these two workflows, HeLa cells were metabolically labeled with AHA for 24 h (pulse) or for about 3 weeks (stable), and half of the material was clicked to either **P-alkyne** or **biotin-alkyne**, for parallel comparisons. Enrichment via PhosID was performed at the peptide level as described herein, while enrichment of biotinylated proteins via streptavidin capture was performed at the protein level. **b** Relative MS intensities of phosphonate-modified or biotinylated peptides. PhosID was 95% selective for phosphonate-labeled peptides that also contained Met → AHA substitutions. Digestion of biotin-streptavidin enriched proteins on the other hand recovered only about 1% of peptides still tagged with biotin. **c** Comparison of identifications from PhosID and biotin-streptavidin workflows. PhosID identified only phosphonate-modified peptides with extremely low background, whereas the biotin-streptavidin approach picked up high background even in HeLa cells not labeled with AHA. Data based on three experimental replicates. A peptide or protein was considered valid when identified in at least two out of three replicates. Protein ID information is provided in Supplementary Data 1. **d** MS intensity correlation between pulse and stable AHA labeling. The extent of POI retrieval from pulse and stable AHA-labeled material were highly similar in both the PhosID and biotin-streptavidin approaches, suggesting that a short pulse of AHA is sufficient to profile the newly synthesized proteome sensitively. $R^2$ values based on the Pearson linear regression model reported. **e** Protein identification overlap between PhosID and Biotin-streptavidin workflows. Data based on 3 experimental replicates. A peptide or protein was considered valid if identified in at least two out of three replicates. Protein ID information is provided in Supplementary Data 3.

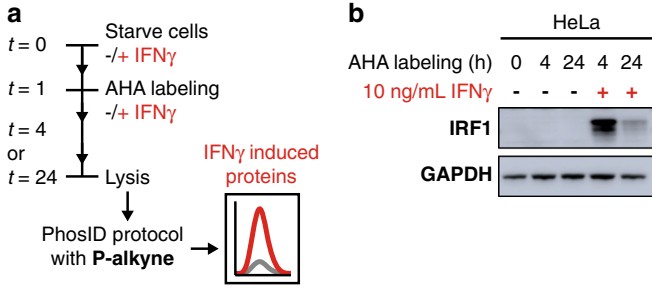

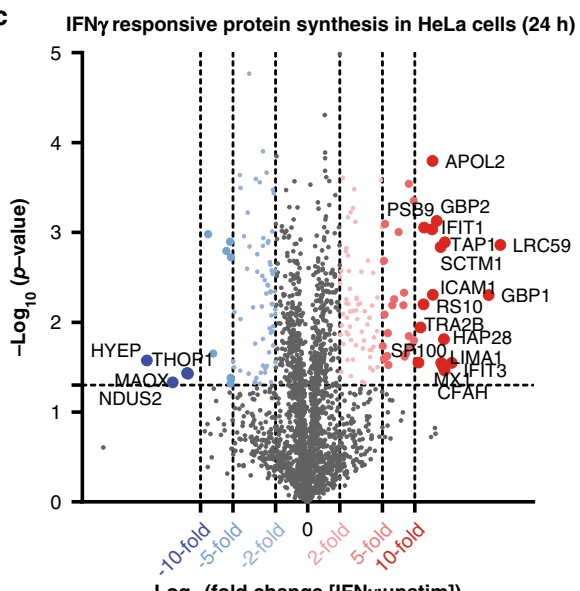

**Fig. 4 Interferon gamma (IFNγ) responsive protein synthesis in HeLa cells. a** IFNγ stimulation timeline. NSPs induced by IFNγ were AHA-labeled, clicked to **P-alkyne**, and enriched using the PhosID protocol. Following IFNγ treatment for either 4 h or 24 h, the abundances of NSPs were determined and normalized against the respective untreated controls at the same time points. **b** Induction of IRF1 in IFNγ treated HeLa cells. Western blot shows a rapid induction of IRF1 at 4 h. By 24 h, IRF1 returned to pre-stimulation levels although downstream signaling and transcription events may still be active. Western blot source data provided in Source data file. **c** Volcano plot of IFNγ-responsive changes in the newly synthesized proteome. Protein targets in red are significantly induced by IFNγ stimulation by 2-, 5- or 10-fold in 24 h, while synthesis of protein targets in blue is significantly suppressed by IFNγ treatment in 24 h. A full list of differentially synthesized proteins is provided in the Supplementary Data 4. Data based on three experimental replicates. A protein was only quantified if identified in at least two out of three replicates.

albeit so far only at the transcript level. Here, for the first time we show their induction or suppression at the protein level. Therefore, with this IFNγ case study, we show not only reliable NSPs isolation using the PhosID protocol, but also demonstrate that systematic retrieval of the newly synthesized proteome that is specific to an exogenous stimulus, is readily achievable with PhosID.

To further test the temporal sensitivity and comparative power of PhosID, we followed new protein synthesis over a shorter duration of 4 h, and also profiled the IFNγ response in a second cell line (Jurkat cells) over the same time points (Supplementary Fig. 9). It has been suggested that Jurkat cells respond differently to IFNγ than HeLa and with different temporal dynamics[48]. By comparing the new protein synthesis in the first 4 h and at 24 h,

we observed a clear two-stage IFNγ response in HeLa cells (Fig. 5a), which was absent in Jurkat cells (Fig. 5b). In HeLa cells, IFNγ specifically triggered early STAT1 signaling (by 4 h) while suppressing the cellular machinery for RNA processing and ribosomal assemblies. Stronger STAT signaling was observed later at 24 h, with a greater diversity of IFNγ-responsive proteins observed and a secondary induction of STAT3 (Fig. 5c, top panel). Jurkat cells on the other hand made most of new proteins in the first 4 h. These are clearly not downstream consequences of STAT1 or part of the pro-inflammatory influence of IFNγ, as STAT1 was only triggered much later at 24 h (Fig. 5c, bottom panel). Furthermore, the different responses we observed in Jurkat are also not due to a shifted response window, as the functional processes mediated by early Jurkat NSPs did not match that of HeLa cells at the later time point. Therefore, IFNγ indeed triggers a more pleiotropic response in Jurkat cells, as hinted at previously[48], and we could conclusively demonstrate this with PhosID in a time-dependent cross-comparison with the Hela response.

## Discussion

In this era of chemical biology, bioorthogonal chemistry has provided a unique and dedicated window to monitor biosynthetic events in cellular systems[3,4,49,50]. Still, there is always room for improvement, and in our view, the poor sensitivity and specificity of biotin-streptavidin affinity retrieval is ultimately limiting the wider and more sensitive application of technologies such as BONCAT, particularly in studying temporal signaling and secretomes[16].

Standard bead-based biotin enrichment workflows proceed at the protein level. This classical approach is somewhat hampered by non-specific enrichments and the difficulty in POI elution. An approach to circumvent this is the use of alkyne-modified agarose-beads for direct click reaction of azide-modified moieties as the sole enrichment step[51]. This simplifies the workflow, but release of captured proteins still requires digestion from the beads[51]. Therefore, biotin-modified peptides, which increase confidence in identification cannot be separated from non-modified peptides. Enrichment at the peptide level is to our knowledge only possible by the use of cleavable linkers[52–54], anti-biotin antibodies[55] or the use of desthiobiotin[56,57], the latter being an affinity-reduced variant of biotin which elutes under milder conditions than biotin. Nonetheless, all of these are still associated with only partial recovery, poor stability during enrichment from complex samples (like whole cell lysates) leading to loss of sensitivity, and contaminations hampering LC-MS/MS identification.

Despite the advances in biotin-based and 'biotin-alike' approaches, there remains a need for a bigger arsenal of simpler, yet higher confidence click chemistry based POI isolation methods, that can be quantitative without additional stable isotope labeling. Driven by these needs to better retrieve, identify and quantify proteins metabolically labeled in live cells, we designed and synthesized novel 'click-able' probes by incorporating a phosphonic acid moiety for direct enrichment and recovery of the peptide containing bioorthogonal modifications.

With the convenient synthesis of **P-alkyne**, **P-azide**, and **P-DBCO** described herein as proof of concept, we demonstrate that the PhosID approach is compatible with diverse flavors of bioorthogonal chemistry (both CuAAC and SPAAC), and are notable additions to the click chemistry toolbox that may be adapted for in-depth analyses of chemically-modified and newly-synthesized proteome subsets. With superior specificity over the biotin-streptavidin enrichment strategy, PhosID also allows sensitive and selective profiling of the actively changing proteome,

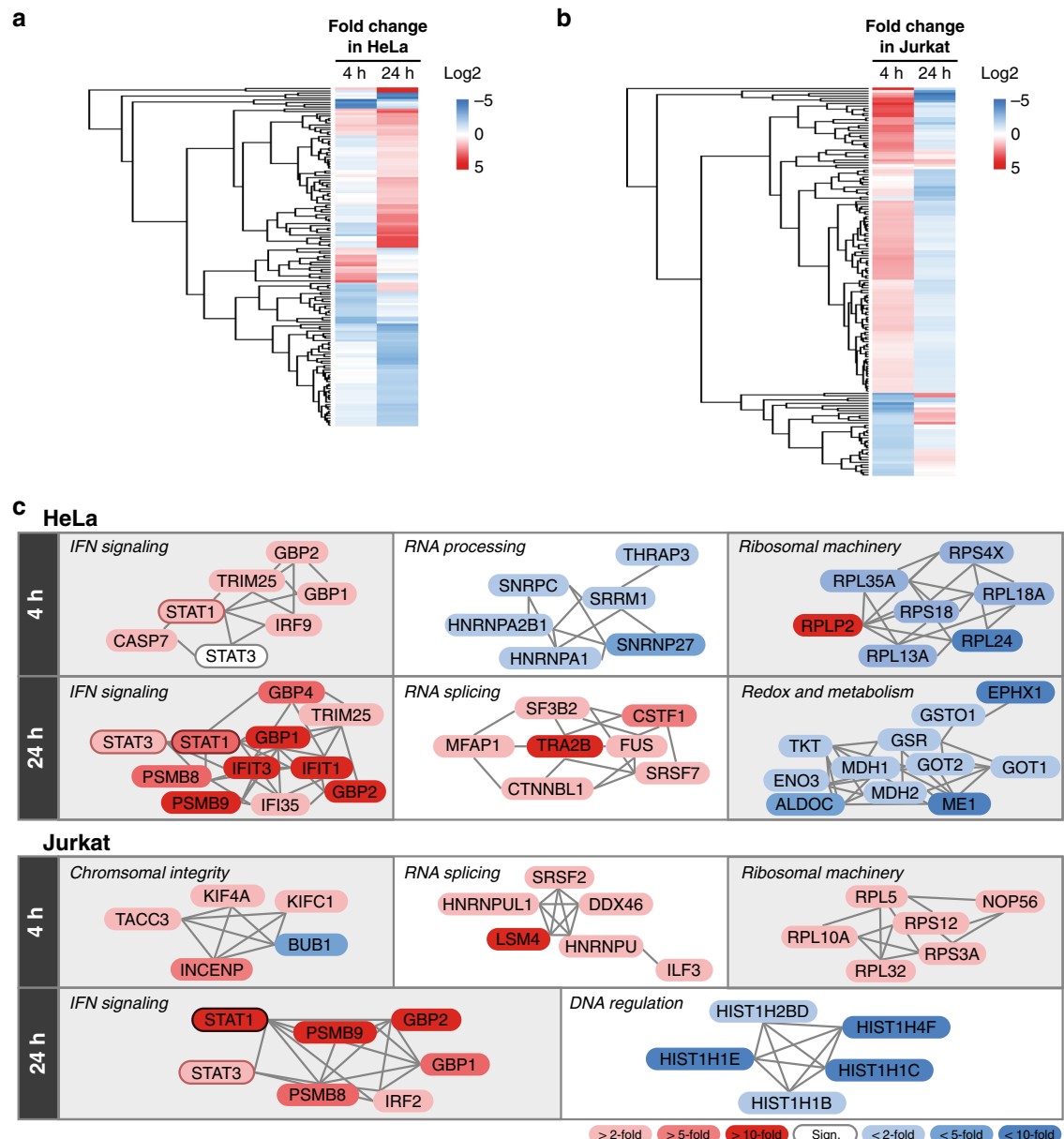

**Fig. 5 Comparison of IFNγ-responsive protein synthesis in HeLa and Jurkat cells. a** Heatmap of temporal IFNγ response in HeLa cells. As evident from the distinct red clusters of proteins, protein synthesis in HeLa cells followed, in general, a two-step induction. Fold change was calculated by the LFQ intensity ratio of IFNγ treated: control samples measured at each time. Data based on three experimental replicates. A protein was only quantified if identified in at least two out of three replicates. **b** Heatmap of temporal IFNγ response in Jurkat cells. As evident from the distinct red clusters of proteins, protein synthesis in HeLa cells followed, in general, a two-step induction. Fold change was calculated by the LFQ intensity ratio of IFNγ treated: control samples measured at each time. Data based on three experimental replicates. A protein was only quantified if identified in at least two out of three replicates. **c** Schematic summary of major temporal differences in IFNγ response. Network relationships were retrieved from STRING.

which we demonstrate here using the **P-alkyne** probe and IFNγ stimulation case studies (Figs. 4 and 5).

Although all the phosphonate-containing probes exemplified in this study were equally specific and efficient in BSA test conditions, these probes still vary in size and chemical affinities. As such, each of these probes could still perform differentially depending on specific biological applications, limits of probe accessibility, and hydrophobicity of different labeling environments. We demonstrate here the feasibility of attaching phosphonate-handles to three such probes, but also envision that many other probe variants may be feasible, to further cater to the diverse flavors of POI isolation by 'click' chemistry.

The strength of PhosID lies firstly in the uncomplicated probe synthesis and very efficient probe retrieval from highly complex peptide mixtures (Table 1). With an one-step mixing reaction that requires no further purification and results in negligible hydrolyzed NHS byproducts, the phosphonate enrichment handle can be introduced into probes for bioorthogonal reactions. This makes it feasible for non-chemists to prepare and functionalize the probes at low costs, but also allows linker lengths and properties to be customized according to experimental needs. In addition, the PhosID protocol adopts widely-used efficient procedures for enzymatic dephosphorylation and IMAC enrichment[37,58], for subsequent retrieval of the phosphonate-handles

**Table 1 Characteristics of the biotin-streptavidin and PhosID workflows.**

|  | Biotin | PhosID |
|---|---|---|
| Ease of probe synthesis | ++ | +++ |
| Ease of modifying linker length and probe properties | + | +++ |
| Probe water solubility | + | +++ |
| Varied formats for handle retrieval | ++ | +++ |
| Sensitivity | + | +++ |
| Specificity | ++ | +++ |
| Elution of labeled peptides | + | +++ |
| Sample processing time | ++ | ++ |
| MS compatibility | ++ | +++ |
| Directly quantitative | + | +++ |
| Multiplexing | + | +++ |
| Reduce sample complexity | + | +++ |
| Cell permeability | + | +++ |
| Application to organisms with high endogenously biotinylated proteins | − | +++ |

Beneficial attributes are represented with (+), while challenges are indicated with (−).

that are intrinsically phosphatase-resistant. This, by design, significantly controls and minimizes the non-specific enrichment background. The choice of the IMAC setup for phosphonate enrichment is quite versatile and can also be performed online in LC systems, in offline spin cartridges or homemade STAGE tips, instead of the liquid handler we used here. The latter however enables automation and improves reproducibility[37]. With the increasing popularity of phospho-peptide enrichment, one of these enrichment formats is likely to be available in the vicinity of every life-science researcher. Collectively, these could make the PhosID protocol readily accessible to most investigators.

In addition to superior sensitivity and specificity over the existing biotin-streptavidin affinity approach (Fig. 3), phosphonate probes have better water solubility characteristics than biotin-based probes and can easily be eluted from the IMAC purification phase, as with phospho-peptides. This also gives PhosID simultaneously more confidence in POI identification and peptide-based quantification, since the same peptide containing the bioorthogonal modification is used for enrichment and quantification. This critically overcomes the chemical artefacts that can be introduced by harsh elution from streptavidin, and preserves MS compatibility in both the qualitative and quantitative sense, as we demonstrate here in a benchmark study using the most classical biotin-streptavidin affinity purification approach in parallel.

In conclusion, through extensive tests detailed in this manuscript and a specific application in the context of studying interferon-responsive newly synthesized proteomes, we have confidently evaluated the conducive properties of phosphonate affinity handles, and outlined their eminent potential to expand the existing toolbox in bioorthogonal chemistry. We envision that the quantitative PhosID approach could be easily adopted in all BONCAT-like applications (alkyne- and azide- modified biomolecules) currently employing the biotin-streptavidin enrichment[14], including profiling protein turnover and the consequences of short signaling pulses, as well as many more contexts of sensitive biological perturbations given the superior sensitivity and low background.

## Methods

**Synthesis of phosphonate-handles**. P-alkyne, P-azide, and P-DBCO (Supplementary Fig. 1a–c) were prepared by conjugating NHS esters with corresponding amine-containing molecules (Fig. 1c). We prepared 250 mM stocks of the commercially available hexynoic acid NHS ester (Alkyne NHS ester, Lumiprobe) as well as azidobutyric acid NHS ester (Azide NHS ester, Lumiprobe) in DMSO. In

addition, dibenzocyclooctyne-N-hydroxysuccinimidyl ester (DBCO-NHS ester, Sigma-Aldrich) was dissolved as 5 mM stock in DMSO. The stock solutions were immediately aliquoted in Eppendorf tubes and stored at −20 °C. The reagents were thawed only once, because the reactive NHS-esters can potentially hydrolyze. A stock solution of 500 mM 2-aminoethyl phosphonic acid (2-AEP, Sigma-Aldrich) was prepared in 1× PBS and adjusted to pH 7.5 using sodium hydroxide. For the in situ generation of 100 μL of 50 mM of P-alkyne or P-azide, 50 μL of 500 mM 2-AEP was incubated with 20 μL of 250 mM Alkyne or Azide NHS ester in a final volume of 100 μL. DBCO-NHS ester was more diluted, because this probe is less water soluble. We added 500 μL of 5 mM DBCO-NHS ester to 500 μL of 25 mM 2-AEP to synthesize P-DBCO. The reactions were performed for at least 2 h at room temperature in the dark rotating resulting in approximately 50 mM stocks of P-alkyne and P-azide, and 2.5 mM stock of P-DBCO. These P-labeled probes were stored at −20 °C.

**Modification of BSA**. Alkyne and Azide NHS ester described in—Synthesis of phosphonate-handles—were used to modify free amine groups (i.e. Lysines) of BSA with alkyne or azide functionalities as bioorthogonal handles on BSA. We prepared 10 mg/mL BSA in 1× PBS (pH 7.5) and a dilution series of the Alkyne and Azide NHS ester in DMSO (Supplementary Table 1). 100 μg of BSA (88 nmol of lysines) was incubated with 22 μL of alkyne or Azide NHS ester stock solution for 2 h at room temperature in 1× PBS (pH 7.5, final volume 200 μL). If the ratio of NHS ester:lysine is not depicted, we have used a ratio of 1:1 (adding 4 mM of NHS stock to BSA). The reaction was quenched by adding 5 μL of 100 mM Tris-HCl (pH 7.5) to the mixture. Residual NHS ester reagents were removed by dialysis against 1× PBS (pH 7.5) using centrifugal units (Amicon MCWO 3 kDa).

**HeLa spike-in experiments**. For the sensitivity and recovery experiments, we equipped BSA with an azide-label using the similar protocol as described in—Modification of BSA—using a ratio of 1:1 of NHS ester to lysine. Modified BSA was added to HeLa lysate using different w/w as indicated in the figures (Fig. 2b and Supplementary Fig. 2) followed by CuAAC as described in "Bioorthogonal chemistry reactions".

**AHA labeling and interferon stimulation**. Cells were starved in DMEM-Met media for 1 h in the absence (pulse and stable experiment) or presence (interferon experiment) of 10 ng/mL Interferon γ (IFN-γ; R&D Systems, MN, USA). In this period, cells would consume >95% of the intracellular methionine pool[41], while accumulating interferon-responsive mRNAs. After methionine starvation, azide-modified methionine analog (L-azido-homoalanine, AHA; Bachem, Bubendorf, Switzerland) was re-introduced into the culture media to a final concentration of 0.1 mM. Cells were harvested after 24 h (pulse experiment) or cultured for ten splittings (about 3 weeks; stable experiment). In case of interferon stimulation, new proteins synthesized with AHA were harvested after 4 h and 24 h from PBS-washed whole cells. Accumulation of interferon-responsive protein IRF1 was verified by western blotting (Fig. 4, Supplementary Fig. 9, Source Data).

**Cell lysis**. Pellets derived from one T75 flask (75 cm²) were resuspended in 500 μL of lysis buffer (1× PBS, pH 7.5, 1% sodiumdeoxycholate (SDC)) containing freshly added 1× protease inhibitor (Roche, EDTA-free). The cells were lysed using a probe sonicator and the supernatant (lysate) was collected by spinning at 15,900 × g for 30 min at 4 °C. The concentration of the proteins was determined by BCA assay and lysates were diluted to 5 μg/μL in 1× PBS (pH 7.5) and stored at −80 °C. 500 μg of lysate was used as input material for both PhosID and biotin enrichment strategies.

**Bioorthogonal chemistry reactions**. The CuAAC or SPAAC reactions were performed for 2 h at room temperature rotating in a total volume of 500 μL with a maximum amount of 500 μg total proteins, containing 2 M Urea (final) in 1× PBS (pH 7.5). CuAAC components were added in the following order (final concentrations are given): 5 mM tris(3-hydroxypropyltriazolylmethyl)amine (THPTA; Lumiprobe) in Milli-Q water (MQ), 2.5 mM CuSO₄ 5·H₂O in MQ, 500 μM of P-alkyne or P-azide (for preparation see "Synthesis of phosphonate-handles") or 500 μM of the commercially available acetylene-PEG₄-Biotin (Biotin-alkyne; Jena Bioscience) in DMSO, and 25 mM sodium ascorbate in MQ. In case of SPAAC, P-DBCO (final concentration of 500 μM) was added to 100 μg of BSA-azide. Bioorthogonal chemistry reactions using P-labels were followed by enrichment using Fe³⁺-IMAC enrichment (PhosID for analysis by LC-MS/MS) and biotin-labeled proteins were enriched using streptavidin (Biotin enrichment for analysis by LC-MS/MS).

**Sample processing for tryptic digestion**. Phosphonate-labeled protein mixtures were dialyzed against 50 mM ammonium bicarbonate (pH 8) and concentrated to 100 μL using centrifugal units (Amicon, MCWO 3 kDa). 100 μL of 8 M of urea in MQ was added to denature these samples. In case of the interferon experiment, proteins were precipitated after CuAAC reaction using chloroform/methanol precipitation. Finally, 500 μg of protein pellet was dissolved in 250 μL of 8 M urea and 250 μL of 50 mM ammonium bicarbonate (pH 8).

**Tryptic digestion**. Proteins were reduced for 30 min at 58 °C by adding dithio-threitol (DTT; prepared fresh, final concentration 2 mM) and alkylated in the dark using iodoacetamide (IAA; prepared fresh, final concentration 4 mM) for 30 min in the dark at room temperature. Residual IAA was quenched by a further 2 mM DTT for at least 30 min. Finally, proteins were digested overnight at 37 °C using Trypsin (1:50, enzyme to protein) and LysC (1:75, enzyme to protein) in a final volume of 500 µL. Digested material was then desalted using $C_{18}$ Seppak.

**Dephosphorylation**. Samples containing human cell lysate (HeLa or Jurkat) were dephosphorylated prior to IMAC enrichment. Desalted peptides were dissolved at a concentration of 2 µg/µL in 1× CutSmart buffer (New England BioLabs; NEB) and 5 units of Alkaline Phosphatase (calf intestinal, CIP from NEB, 10,000 units/mL) was added per 100 µg of protein material. After dephosphorylation overnight at 37 °C with shaking, peptides were desalted using $C_{18}$ Seppak.

**Automated Fe(III)-IMAC workflow**. Enrichment protocol was performed simi-larly as described by Steigenberger et al.[32] following the procedures also described by Post, et al.[37] In detail, PhosID-labeled peptides were enriched using Fe(III)-NTA 5 µL (Agilent technologies) in an automated fashion by the AssayMAP Bravo Platform (Agilent Technologies). Fe(III)-NTA cartridges were primed at a flow rate of 100 µL/min with 250 µL of priming buffer (0.1% TFA, 99.9% ACN) and equi-librated at a flow rate of 50 µL/min with 250 µL of loading buffer (0.1% TFA, 80% ACN). The flow through was collected into a separate plate. Dried samples were dissolved in 200 µL of loading buffer and loaded at a flow rate of 5 µL/min onto the cartridge. Columns were washed with 250 µL of loading buffer at a flow rate of 20 µL/min, and the PhosID-labeled peptides were eluted with 35 µL of ammonia (10%) at a flow rate of 5 µL/min directly into 35 µL of formic acid (10%).

**Biotin enrichment samples preparation for LC-MS/MS**. Biotin-labeled samples were dialyzed against 1× PBS (pH 7.5) using centrifugal units (Amicon, MCWO 3 kDa). We added 100 µL of Streptavidin Agarose slurry (Merckmillipore), 2 M urea (final concentration), 1% SDC (final w/v), and 1× protease inhibitor. After incubating the samples overnight at 4 °C, the flow through was discarded and the beads were washed three times with 1 mL of the following buffers: (i) 1% SDC and 5 mM ethylenediaminetetraacetic acid (EDTA) in 1× PBS (pH 7.5), (ii) 6 M urea, (iii) 1× PBS (pH 7.5). Beads were resuspended in 100 µL of 50 mM ammonium bicarbonate and 100 µL of 8 M urea. Proteins were then reduced, alkylated, quenched, and digested using the similar protocol as described under PhosID for analysis by LC-MS/MS. After overnight digestion at 37 °C, peptides were eluted from the beads and desalted using $C_{18}$ Seppak before LC-MS/MS.

**LC-MS/MS**. Briefly, samples were analyzed on an UHPLC 1290 system (Agilent Technologies; Santa Clara, Ca) coupled to an Orbitrap Fusion (Thermo Scientific, San Jose, Ca) or an Orbitrap Q Exactive Pluss mass spectrometer (Thermo Scientific, San Jose, Ca), in data-dependent acquisition mode, largely as described previously[32]. Peptides were trapped on a 2 cm × 100 µm Reprosil C18 precolumn (3 µm) for 10 min in solvent A (0.1 % v/v formic acid in water) and then separated on an analytical column (Agilent Poroshell EC-C18, 2.7 µm, 50 cm × 75 µm). For the analysis of samples containing BSA without HeLa lysate, we used a gradient of 0–10% solvent B (0.1 % v/v formic acid in 80% acetonitrile (ACN)) in 5 min, 10–44% in 30 min, 44–100% in 10 min, and 100% solvent B for 5 min. Samples derived from cell lysate were analyzed using a gradient of 0–10% solvent B in 5 min, 10–38% in 95 min, 38–100% in 3 min, and 100% solvent B for 1 min. Finally, the system was equilibrated back to 100% solvent A for 11 min. In all cases, the flow was passively split to ~200 nL/min. MS1 was performed at a resolution of 60,000, between 375 and 1600 $m/z$ after accumulation to $3 \times 10^6$ ions in a max-imum injection time of 20 ms. Top 15 most intense precursors were fragmented with NCE 27 and 16 s dynamic exclusion time. HCD fragmentation was performed on precursors at a resolution of 30,000, between 200 and 2000 $m/z$ after accu-mulation to $1 \times 10^5$ ions in a maximum injection time of 50 ms.

**Data analysis and visualization**. Raw files were processed using MaxQuant ver-sion 1.6.7 and the Andromeda search engine[59], against either the BSA (downloaded from uniprot) or the human (20431 entries) Swissprot database (version Sep 2019). Enzyme specificity was set to Trypsin, and up to two missed cleavages were allowed. Cysteine carbamidomethylation was set to fixed modification, whereas variable modifications of methionine oxidation and protein N-terminal acetylation were allowed, together with methionine to AHA substitution ($C_{-1}H_{-3}N_3S_{-1}$), phosphorylation (STY), or one of the other modifications: lysine labeled with azide ($C_4H_5N_3O_1$), alkyne ($C_6H_6O_1$), CuAAC with one of the P-labels ($C_{12}H_{19}N_4O_5P_1$), SPAAC with the **P-DBCO** ($C_{25}H_{26}N_5O_6P_1$), reactions with one of the side pro-ducts originating from the synthesis of the P-labels (Supplementary Fig. 1d-f; $C_{10}H_{13}N_3O_3$ or $C_{23}H_{20}N_4O_4$); methionine to phosphonate substitution ($C_7H_{11}N_4O_4P_1S_{-1}$). False discovery rate (FDR) was restricted to 1% in both protein and peptide identification. Label-free quantification (LFQ) was performed for BSA experiments with match between runs enabled, in separate groups for the input, flow through, and enriched fractions. iBAQ was performed for the experiment comparing phosphonate and biotin enrichment with match between runs enabled in six separate groups for phosphonate and biotin (HeLa, AHA pulse labeling, AHA stable labeling). For quantitative comparisons of the interferon experiment, LFQ was performed with "match between runs" enabled, in separate parameter groups for 4 h and 24 h measurements.

**Data filtering parameters**. After MaxQuant searches, BSA experiments were fil-tered for non-reverse, and the leading razor protein being P02769, Score more than 20 and a PEP score of below 0.05. Both the experiment comparing phosphonate and biotin, and the interferon experiment were filtered for non-reverse, potential contaminants were discarded. We did not include filtering for the Score or PEP, because the Score for phosphonate-labeled proteins will be inherently low as we mostly have 1–2 peptides per protein. Data were visualized using GraphPad PRISM 8.

**Reporting summary**. Further information on research design is available in the Nature Research Reporting Summary linked to this article.

## Data availability
The mass spectrometry data have been deposited to the ProteomeXchange Consortium via the PRIDE partner repository with the data set identifier PXD018272. The source data underlying Figs. 2a and 4, Supplementary Figs. 2, 3, 5, 6, and 9 are provided as Source data. Source data are provided with this paper.

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

## Acknowledgements

We acknowledge support from the Dutch Research Council (NWO) funding the Netherlands Proteomics Centre through the X-omics Road Map program (project 184.034.019) and the EU Horizon 2020 program INFRAIA project Epic-XS (Project 823839).

## Author contributions

A.J.R.H. and W.W. conceived the study and designed the approach. Supported by B.S., F.K. developed and synthesized the reagents and performed the experiments. W.W. performed the cell labeling and F.K. processed the samples further. F.K., W.W., and B.S. performed data analysis. All authors together wrote and edited the paper.

## Competing interests

The authors declare no competing interests.
