## [Peer Review File · Nature Communications]

Reviewers' comments:

Reviewer #1 (Remarks to the Author):

In this work, Kleinpenning et al. report a new strategy termed PhosID to purify proteins from complex mixtures using bioorthogonal noncanonical amino acid tagging (BONCAT). By using click chemistry to attach phosphonate handles onto metabolically incorporated azidohomoalanine residues, the authors show that they can effectively capture and release peptides containing the modified noncanonical amino acids. They synthesize the phosphonate tags with relative ease for both copper catalyzed and strain promoted azide alkyne cycloaddition, and they demonstrate that enrichments at the peptide level using PhosID have lower background signal than enriching at the protein level with biotin, which is a useful benefit. They demonstrate the utility of PhosID in an experiment that reveals temporal differences in proteome response between HeLa and Jurkat cells upon IFN γ stimulation. Overall, this is a well written manuscript and an important addition to the bioorthogonal chemistry toolkit. That said, there are enrichments at the peptide level using biotin (with anti-biotin antibodies) or desthiobiotin that, in theory, could accomplish the same goal. There is no doubt that PhosID is an improvement of protein-level enrichment with biotin followed by on-bead digestion, as their data shows here, but it would have been useful to see a straightforward comparison between these peptide-level enrichments and PhosID. I recognize that further experiments are difficult to conduct given current events worldwide, so I suggest that the authors at least discuss these other options in the manuscript at all points when comparing biotin enrichments to PhosID. I also have a few other comments to address, as indicated below. Even so, I believe this is a strong manuscript that can be ready for acceptance upon mostly minor revisions.

- Discussion of desthiobiotin as a more transient/elutable version of the biotin/streptavidin interaction is necessary and warranted. I recognize that current events make it difficult to ask for more experiments, although a comparison of PhosID with desthiobiotin-based enrichment would be very interesting. That said, if the authors have any data on this on hand already, it would be a welcomed addition. One could imagine using desthiobiotin-NHS and desthiobiotin-alkyne to accomplish the same goal described here, even at the peptide enrichment level. Sentences such as "Conceptually, PhosID and biotin-streptavidin are distinct as PhosID enriches at the peptide level whereas the latter captures at the protein level", page 5 would be nullified when consider desthiobiotin, which could enrich at the peptide level. Such sentences would also be nullified by the peptide-level biotin enrichment protocols cited in the introduction using anti-biotin antibodies. Quantitation comparisons would also be more straightforward using desthiobiotin or anti-biotin antibodies for enrichment at the peptide level. PhosID is a useful addition to the chemical biology toolbox regardless, but the caveat of desthiobiotin as an alternative should be addressed (benefits of not needing phosphatase treatment, etc).
- Can the authors comment on the number of sialylated and phosphorylated glycopeptides that were enriched (a known phenomenon in IMAC enrichments)? This could be done for experiments where AHA was incorporated and when it was not (the background experiments in Figure 3B and Supplementary Figure 4). A simple count of spectra containing signature oxonium ions would be sufficient to comment on how many spectra in the enriched sample may be from species not from the PhosID specific peptides, given that phosphopeptides are largely eliminated (Supplementary Table 2). It would be valuable to know the number of interfering species that one could expect in a PhosID experiment, and if PNGaseF treatment should be added to the workflow along with phosphatase treatment.
- Along the lines with the glycopeptide enrichment comment above, can the authors provide identifications of peptides/proteins that were seen in the background enrichment experiments where no AHA was included. I am mostly interested in identifications from the background PhosID enrichment – are peptides bound by IMAC here mostly glycosylated, or are there other proteins that could be filtered out of IMAC-enriched datasets? They provide the number of non-specific background proteins from the unlabeled HeLa lysate for the biotin enrichment (> 2000). Could those identifications

be provided in as supplemental data along with the PhosID background IDs? Is digested streptavidin the major component of the total peptide mass from this background experiment, or are there other “sticky” background proteins that contribute substantially to the signal seen? A quick note about the abundance of streptavidin peptides in the background would be a valuable addition to the text.

- It would also be interesting to have the protein identifications from Figure 3E as supplemental data to compare.
- Instead of static cut-offs used for significant changes in the IFN γ (>2-fold, $p < 0.05$), I recommend the authors perform a multiple testing correction for significance, as is available in Perseus. It may not change the results much, but it is a more sound way of calculating significant changes.
- Minor comment: if you are going to cite the BioSITE work, it is probably good to include the Carr et al. work on anti-biotin antibodies, as well: <https://www.nature.com/articles/nmeth.4465>
- Minor comment, could the authors make the black borders for colors in figures less thick (or eliminate them entirely)? This would make it easier to see small percentages that are supposed to be a given color but currently look black because of the border lines.

Reviewer #2 (Remarks to the Author):

The total cellular proteome can nowadays be measured by MS with quite some success in depth and precision. However, there is still need for specific technologies focusing on selected smaller subsets of proteins. The manuscript by Kleinpenning et al. introduced three new phosphonate-handles. These probes selectively react with azide or alkyne containing biomolecules. Then the phosphonate-containing biomolecules can be enriched with the widely used immobilized metal affinity chromatography (IMAC) in proteomics. The authors investigated newly synthesized proteins to benchmark these probes. Substitution of the Met in the cell culture medium using its analog AHA enables the incorporation of azides into newly synthesized proteins. The phosphonate-containing probes selectively react with the azides. Then IMAC was used to enrich the phosphonate-containing peptides for mass spectrometry analysis. The enrichment at peptide level is attractive in consideration of minimizing contaminants. The results showed these probe work. However, there are a few areas that should use elaboration/clarification to improve the manuscript, especially considering the scope and broad readership of Nature Communications.

Major concerns:

1. The title and abstract strongly indicate the probes are developed to investigate newly synthesized proteins. However, the introduction and discussion did not mention newly synthesized proteins, did not talk about previous tools (e.g. AHA)/methods for newly synthesized proteins. If the authors intend to present a new method/strategy for newly synthesized proteins, related research should be covered in the introduction and discussion.
2. Theoretically the probes can be used to enrich any azide- or alkyne-containing biomolecules. If the investigation of newly synthesized proteins is just for benchmarking one of the applications of the probes, it's better to clarify it in the manuscript and discuss other applications as well.
3. As proof of concept, the authors showed that these probes work on functionalized BSA. But in HeLa experiment and INF experiment, it seems only one probe was demonstrated. It's unclear how the other two probes perform in a real case scenario.
4. What is the difference and similarity among these three probes in respective of performance,

reactivity, specificity, stability, cost, etc? Are they biased towards a specific subset of peptides in cell line samples? How about the overlap between the identified peptides using different probes? It's also confusing which probe is used in an experiment in the figures and text. Adding a couple words in the figures showing workflows should clarify it.

5. In comparison with existing strategies, the authors adopted a biotin-based strategy. There is another widely used agarose-based strategy (alkyne agarose beads, as used in Ref14). The alkyne agarose strategy is supposed to perform better than the biotin-based strategy. What's the advantages and disadvantage of PhosID in comparison of the alkyne agarose strategy? Have the authors done any experiments to compare them?

6. A reference and a couple of sentences explaining "click-chemistry" in the introduction will be helpful considering the wide readership of Nature Communications

7. The authors said "In addition, only a negligible amount (less than 2%) of the peptides were found to be labeled with a partially hydrolyzed probe synthesis byproduct (Supplementary Figure 2)." How was the 2% calculated?

8. In Fig3D, the author said "In addition, PhosID does not bias for longer AHA labeling times, as pulse-AHA and stable-AHA labeled samples were also found to be highly correlated (coefficient 0.99) (Figure 3D)." It's not clear what bias the authors are talking about here. The number of peptides or proteins? If the authors are talking about number, then the numbers in each group should also be specified. Should cells generate more newly synthesized protein in stable-AHA labeled samples (higher numbers of NSPs and higher abundance for a given NSP)?

9. The authors studied the newly synthesized protein in both HeLa and Jurkat cell lines to benchmark one of the probes.

a. Fig 4 only shows the HeLa results. Results in Jurkat should also be described in the same way.

b. A table summarizing the results in Fig 4c will be helpful. For example, how many peptides/proteins quantified, how many went up/down significantly.

Minor points:

1. The name "PhosID" is a little confusing. The first impression is this is a strategy for studying phosphorylation.

2. Showing the AHA structure in Fig 3a can help with the understanding.

3. Fig1C, the illustration of the PhosID is not clear enough. For example, under the leftmost arrow, it says "Digest", but the arrow points to a full protein. In addition, a 3-D protein structure is confusing and misleading if the protein is already dissolved in lysis buffer.

4. In the Methods, the last sentence in the "HeLa spike-in experiments" paragraph, "supplementary Figure X" should be clarified.

5. In the Methods, in the 4th row in the "Cell culture. AHA labeling and interferon stimulation" paragraph, clarify "PMID: 29669735".

6. In the methods, in the "Bioorthogonal chemistry reactions" paragraph, explain "MQ"

7. In the methods, in the "PhosID sample preparation for LC-MS/MS" paragraph

a. pH is missing for many buffers. For example, 50 mM ABC and 8 M Urea. Also specify in what solution the ABC and Urea Buffer are prepared.

b. Describe the final conc. of DTT and IAM.

c. Automated Fe(III)-IMAC workflow, provide essential description here instead of just citing two

papers.

8. In the methods, in the "LC-MS/MS" paragraph, some key parameters are missing or incomplete. For example, "dynamic exclusion duration of 16 sec" did not specify the times. Collision energy is also missing.

9. Fig3A, fix the typo "Dephosporlyate".

10. Fig3D, define the correlation, pearson or spearman? R or R-square? What is LC-MS intensity? Is it peptide or protein intensity? Or just use iBAQ intensity instead of LC-MS intensity since iBAQ intensity is explained in the methods.

11. In the discussion it said the experiment in Fig4 used one (P-alkyne) of the probes. This should also be clearly indicated in Fig4a instead of using an ambiguous saying "PhosID protocol".

12. Fig S5, what's the "log" in the y-axis label?

13. Fig S7, define the correlation. R or R-squared? Pearson or spearman?

14. Fig 4c, x axis label is missing. Texts are overlapping with each other.

15. Table S4

a. In the "content" tab, "T.test: t.test based on three technical replicates, two-tailed, paired". Is it p value or t statistics? Or log-transformed p value?

b. Fix the header of the last tab, they don't match the content.

16. Fig2A, define the black part (labeled by the by-product?)

17. Fig5a and Fig5b, are ratios capped at 5 and -5?

Reviewers' comments: (our point to point responses are in blue)

Reviewer #1 (Remarks to the Author):

In this work, Kleinpenning et al. report a new strategy termed PhosID to purify proteins from complex mixtures using bioorthogonal noncanonical amino acid tagging (BONCAT). By using click chemistry to attach phosphonate handles onto metabolically incorporated azidohomoalanine residues, the authors show that they can effectively capture and release peptides containing the modified noncanonical amino acids. They synthesize the phosphonate tags with relative ease for both copper catalyzed and strain promoted azide alkyne cycloaddition, and they demonstrate that enrichments at the peptide level using PhosID have lower background signal than enriching at the protein level with biotin, which is a useful benefit. They demonstrate the utility of PhosID in an experiment that reveals temporal differences in proteome response between HeLa and Jurkat cells upon IFN γ stimulation. Overall, this is a well written manuscript and an important addition to the bioorthogonal chemistry toolkit.

That said, there are enrichments at the peptide level using biotin (with anti-biotin antibodies) or desthiobiotin that, in theory, could accomplish the same goal. There is no doubt that PhosID is an improvement of protein-level enrichment with biotin followed by on-bead digestion, as their data shows here, but it would have been useful to see a straightforward comparison between these peptide-level enrichments and PhosID. I recognize that further experiments are difficult to conduct given current events worldwide, so I suggest that the authors at least discuss these other options in the manuscript at all points when comparing biotin enrichments to PhosID. I also have a few other comments to address, as indicated below. Even so, I believe this is a strong manuscript that can be ready for acceptance upon mostly minor revisions.

- Discussion of desthiobiotin as a more transient/elutable version of the biotin/streptavidin interaction is necessary and warranted. I recognize that current events make it difficult to ask for more experiments, although a comparison of PhosID with desthiobiotin-based enrichment would be very interesting. That said, if the authors have any data on this on hand already, it would be a welcomed addition. One could imagine using desthiobiotin-NHS and desthiobiotin-alkyne to accomplish the same goal described here, even at the peptide enrichment level. Sentences such as “Conceptually, PhosID and biotin-streptavidin are distinct as PhosID enriches at the peptide level whereas the latter captures at the protein level”, page 5 would be nullified when considering desthiobiotin, which could enrich at the peptide level. Such sentences would also be nullified by the peptide-level biotin enrichment protocols cited in the introduction using anti-biotin antibodies. Quantitation comparisons would also be more straightforward using desthiobiotin or anti-biotin antibodies for enrichment at the peptide level. PhosID is a useful addition to the chemical biology toolbox regardless, but the caveat of desthiobiotin as an alternative should be addressed (benefits of not needing phosphatase treatment, etc).

We agree with the reviewers that desthiobiotin and other variants should be better discussed in the manuscript, and have added a section in paragraph 2 of discussion (page 9) to highlight their specific advantages. Unfortunately we do not have desthiobiotin data to make fair comparisons, but argue in the revised manuscript that disadvantages, remain with these approaches, that can be addressed by using PhosID.

We also delete the sentence “Conceptually...”, and reword the paragraph opening for the “Comparison of PhosID with biotin enrichment” Results sub-section (page 6).

- Can the authors comment on the number of sialylated and phosphorylated glycopeptides that were enriched (a known phenomenon in IMAC enrichments)? This could be done for experiments where AHA was incorporated and when it was not (the background experiments in Figure 3B and Supplementary Figure 4). A simple count of spectra containing signature oxonium ions would be sufficient to comment on how many spectra in the enriched sample may be from species not from the PhosID specific peptides, given that phosphopeptides are largely eliminated (Supplementary Table 2). It would be valuable to know the number of interfering species that one could expect in a PhosID experiment, and if PNGaseF treatment should be added to the workflow along with phosphatase treatment.

We agree with the reviewer that TiO₂-based phospho-enrichment procedures may co-enrich sialylated glycopeptides. However, we performed phospho-enrichment here with Fe-IMAC, which is less prone to co-isolation of glycopeptides, as we and others have demonstrated before experimentally.

In our previous work using the same Fe-IMAC procedures for characterization of phosphorylated glycopeptides, we noted almost complete absence of sialylated glycopeptides in Fe-IMAC enriched material (PMID: 30237200; PMID: 31287300), which was in contrast to known Titanium-based methods. For instance in Caval et al., PMID: 30237200, we found less than 60 scans containing oxonium ions corresponding to complex sialylated glycopeptides, which is a negligible amount when considering 50 000 scans on average per raw file. Using the same tool, we counted spectra containing signature oxonium ions (204 and 274.292 respectively), and found on average around 20 scans containing oxonium ions corresponding to complex sialylated glycopeptides per raw file, which is similarly negligible in our opinion. Consequently, we think PNGaseF co-treatment is not necessary, and unlikely to substantially improve identifications further.

- Along the lines with the glycopeptide enrichment comment above, can the authors provide identifications of peptides/proteins that were seen in the background enrichment experiments where no AHA was included. I am mostly interested in identifications from the background PhosID enrichment – are peptides bound by IMAC here mostly glycosylated, or are there other proteins that could be filtered out of IMAC-enriched datasets? They provide the number of non-specific background proteins from the unlabeled HeLa lysate for the biotin enrichment (> 2000). Could those identifications be provided in as supplemental data along with the PhosID background IDs?

We now provide the background identification list in the amended submission (Supplementary Table 4). In addition, we also checked that these background identification from the biotin-based enrichment do not enrich for specific cellular compartments; about 10% of these function in protein, lipid and carbohydrate metabolism, which is in the same range as what would be expected from shotgun analysis of a total cell lysate; and only 1 protein has been reported previously to be involved in biotin metabolism (Uniprot Accession P05165). Taken together, we still think these background proteins from (unlabelled HeLa lysate) biotin enrichment are indeed just non-specific binding.

Is digested streptavidin the major component of the total peptide mass from this background experiment, or are there other “sticky” background proteins that contribute substantially to the signal seen? A quick note about the abundance of streptavidin peptides in the background would be a valuable addition to the text.

Streptavidin is part of the Maxquant “contaminants” list, and therefore was filtered out of our quantitative data at the very beginning of analysis. Thus, digested streptavidin is not part of the identified non-specific background that we report in Figure 3E.

Nonetheless, on query of the reviewer, we went back to the “contaminants” we filtered out, and verified that also very few streptavidin peptides were detected there. This is now shown in Supplementary Table 5, where <10 streptavidin peptides were identified as “contaminants” per condition. The intensities of these streptavidin peptides are also in the low range of <0.5% to 3% out of the total intensity. We now also clarify this in-text at the end of the Results subsection “Comparison of PhosID with biotin enrichment” (page 6).

- It would also be interesting to have the protein identifications from Figure 3E as supplemental data to compare.

This is now provided as Supplementary Table 6.

- Instead of static cut-offs used for significant changes in the IFN γ (>2-fold, $p < 0.05$), I recommend the authors perform a multiple testing correction for significance, as is available in Perseus. It may not change the results much, but it is a more sound way of calculating significant changes.

We now provide the FDR-corrected p-value as an additional column in Supplementary Table 7. This indeed did not change the results much.

- Minor comment: if you are going to cite the BioSITE work, it is probably good to include the Carr et al. work on anti-biotin antibodies, as well: <https://www.nature.com/articles/nmeth.4465>

We have now added the reference, number 55.

- Minor comment, could the authors make the black borders for colors in figures less thick (or eliminate them entirely)? This would make it easier to see small percentages that are supposed to be a given color but currently look black because of the border lines.

We thank the reviewer for pointing this out. We have amended the figures, in particular, please see Figure 2.

Reviewer #2 (Remarks to the Author):

The total cellular proteome can nowadays be measured by MS with quite some success in depth and precision. However, there is still need for specific technologies focusing on selected smaller subsets of proteins. The manuscript by Kleinpenning et al. introduced three new phosphonate-handles. These probes selectively react with azide or alkyne containing biomolecules. Then the phosphonate-containing biomolecules can be enriched with the widely used immobilized metal affinity chromatography (IMAC) in proteomics. The authors investigated newly synthesized proteins to benchmark these probes. Substitution of the Met in the cell culture medium using its analog AHA enables the incorporation of azides into newly synthesized proteins. The phosphonate-containing probes selectively react with the azides. Then IMAC was used to enrich the phosphonate-containing peptides for mass spectrometry analysis. The enrichment at peptide level is attractive in consideration of minimizing contaminants. The results showed these probe work. However, there are a few areas that should use elaboration/clarification to improve the manuscript, especially considering the scope and broad readership of Nature Communications.

Major concerns:

1. The title and abstract strongly indicate the probes are developed to investigate newly synthesized proteins. However, the introduction and discussion did not mention newly synthesized proteins, did not talk about previous tools (e.g. AHA)/methods for newly synthesized proteins. If the authors intend to present a new method/strategy for newly synthesized proteins, related research should be covered in the introduction and discussion.

On advice from the reviewer, we now expand on this in paragraph 2 of the introduction (page 3), although also most of the papers we cited in the introduction already focus on the analysis of newly synthesized proteins as applications.

2. Theoretically the probes can be used to enrich any azide- or alkyne-containing biomolecules. If the investigation of newly synthesized proteins is just for benchmarking one of the applications of the probes, it's better to clarify it in the manuscript and discuss other applications as well.

We now specify the scope of application in this manuscript to "newly synthesized proteins", in the last paragraph of the introduction (page 3-4). This may indeed focus the manuscript better on utility of phosphonate handles in the retrieval of newly synthesized proteomes, which is the bulk of our content, and agrees better with our title. We thank the reviewer for this suggestion.

In addition, we also still list potential wider application of phosphonate handles in other contexts at the end of the Discussion (page 10).

3. As proof of concept, the authors showed that these probes work on functionalized BSA. But in HeLa experiment and INF experiment, it seems only one probe was demonstrated. It's unclear how the other two probes perform in a real case scenario.

The main goal of this manuscript is to introduce the phosphonate handle, which is the central novelty here, and not really to benchmark comparative efficiency of different probes. In theory, the phosphonate handle can be added to any existing probe as long as it still can sterically accommodate the small enrichment handle. The wide selection of all possible probes meant we could only

showcase, as proof of principle, the addition of phosphonate to three most popular probes used by investigators in the field. Despite DBCO, alkyne and azide being quite different chemical moieties, we almost did not observe any difference in reaction specificity of these phosphonate modified probes on functionalised BSA. We think this already demonstrates the robust applicability of phosphonate handles in general.

While cross-comparing the efficiencies of different probes containing the same phosphonate handle is not an explicit goal in this work, we also do not expect to see larger differences than what is intrinsic to the reactive chemistry of these probes without phosphonate-tagging.

4. What is the difference and similarity among these three probes in respective of performance, reactivity, specificity, stability, cost, etc? Are they biased towards a specific subset of peptides in cell line samples? How about the overlap between the identified peptides using different probes? It's also confusing which probe is used in an experiment in the figures and text. Adding a couple words in the figures showing workflows should clarify it.

We now specify the probe used with the PhosID protocol on Figures 3 and 4. However, as addressed in question 3, comparison of different phosphonate-containing probes was not the goal of our investigation. We demonstrate the possibility to add phosphonate handles to 3 widely used existing probes, but it would be confounding to compare different phosphonate-containing probes against each other, as potential differences can also be intrinsic to the reactive chemistry of alkyne, azide and DBCO, instead of our phosphonate handle. We think such comparisons will be hard to interpret, and do not contribute to the potential wider applicability that we try to advocate here.

5. In comparison with existing strategies, the authors adopted a biotin-based strategy. There is another widely used agarose-based strategy (alkyne agarose beads, as used in Ref14). The alkyne agarose strategy is supposed to perform better than the biotin-based strategy. What's the advantages and disadvantage of PhosID in comparison of the alkyne agarose strategy? Have the authors done any experiments to compare them?

We think alkyne-agarose beads would perform at similar levels as streptavidin-agarose beads, since both are bead-based approaches, that suffer from the same non-specific binding to beads, and incur the similar steric hindrance in capturing reactive substrate, which results from bead immobilisation. Although there is some potential to only partially recover peptides from streptavidin-agarose than digestion (harsh conditions), both streptavidin- and alkyne-agarose are not readily and reproducibly elute-able. Therefore, with these considerations, we did not envision how alkyne-agarose would be superior to streptavidin-agarose, to warrant an additional direct comparison to enrichment with the phosphonate handle.

Conversely, the phosphonate handle is much smaller without the bead support (in the PhosID strategy), and eliminating the use of beads also gets rid of the 'sticky' background at the same time. Elution from Fe-IMAC is also routinely achievable. With all the ways we can rationalise potential improvement in performance, we did not see how PhosID would do better than streptavidin-agarose, but worse than alkyne-agarose. We therefore did not include alkyne-agarose in the experimental design stage.

6. A reference and a couple of sentences explaining "click-chemistry" in the introduction will be helpful considering the wide readership of Nature Communications

We agree with the reviewer general explanations on "click chemistry" are needed in the introduction. We now expand on this in paragraph 2 of the introduction (page 3).

7. The authors said "In addition, only a negligible amount (less than 2%) of the peptides were found to be labeled with a partially hydrolyzed probe synthesis byproduct (Supplementary Figure 2)." How was the 2% calculated?

We now clarify this further in the Supplementary Figure 1 legend, with "Less than 2.5% of the peptides were found to be reacted with the byproducts; 100% is the sum of unmodified, azide- or alkyne-labeled peptides, phosphonate-labeled peptides and byproducts". The base for calculating this percentage is thus all the peptides, including labeled, unlabeled and byproducts.

8. In Fig3D, the author said "In addition, PhosID does not bias for longer AHA labeling times, as pulse-AHA and stable-AHA labeled samples were also found to be highly correlated (coefficient 0.99) (Figure 3D)." It's not clear what bias the authors are talking about here. The number of peptides or proteins? If the authors are talking about number, then the numbers in each group should also be specified. Should cells generate more newly synthesized protein in stable-AHA labeled samples (higher numbers of NSPs and higher abundance for a given NSP)?

Longer labeling times will increase the copy number of newly synthesized proteins containing AHA, and hence improve sensitivity and quantitative accuracy (accumulation of newly synthesized proteins is a function of time). Here we mean that even in short labeling times, the quantitative correlation is still very high to the values of stable-AHA, suggesting that even quantitation from shorter pulse labeling experiments are just as reliable as longer labeling experiments.

9. The authors studied the newly synthesized protein in both HeLa and Jurkat cell lines to benchmark one of the probes.

a. Fig 4 only shows the HeLa results. Results in Jurkat should also be described in the same way.

As the changes induced by IFN gamma in HeLa and Jurkat cells overlap partially, we did not show the Jurkat data as a separate figure in the first instance. This is because doing so will feature some similar data, also in similar format; even the sub-figure A would be duplicated, since the treatment sequence and workflow is also identical.

What we deemed more important was to show the temporal differences in response between the HeLa and Jurkat lines. This, we thought, is best captured as a heatmap over time, which we show in Figure 5 in the original submission.

The Jurkat IRF1 western blot from is shown in Supp figure 9. The Jurkat dataset was also fully listed in Supplementary Table 4, with all the differentially regulated proteins at 4h and 24h in different tabs, similarly to the HeLa dataset within the same excel table.

Therefore, we argue the Jurkat data are fully described.

b. A table summarizing the results in Fig 4c will be helpful. For example, how many peptides/proteins quantified, how many went up/down significantly.

We now provide these required summaries in a new tab added to the existing Supplementary Table 7.

Minor points:

1. The name "PhosID" is a little confusing. The first impression is this is a strategy for studying phosphorylation.

We did consider other names (e.g. PhosphonateID) before choosing to shorten it to 'PhosID' which we thought was the most appropriate. If the editor feels also strongly about this, we could change the name but find it difficult to come with a better alternative.

2. Showing the AHA structure in Fig 3a can help with the understanding.

The AHA structure is now added in Figure 3A. Thank you for the suggestion.

3. Fig1C, the illustration of the PhosID is not clear enough. For example, under the leftmost arrow, it says "Digest", but the arrow points to a full protein. In addition, a 3-D protein structure is confusing and misleading if the protein is already dissolved in lysis buffer.

We indeed agree with the reviewer, thank you for pointing this out. In the amended Figure 1C, we moved the steps 3 and 4 to go with the second arrow, to avoid the confusion.

4. In the Methods, the last sentence in the "Hela spike-in experiments" paragraph, "supplementary Figure X" should be clarified.

The figure reference should have been "supplementary Figure 2". This is now rectified in the updated version (page 15).

5. In the Methods, in the 4th row in the "Cell culture. AHA labeling and interferon stimulation" paragraph, clarify "PMID: 29669735".

The reference should have been added in the reference list. This is now rectified in the latest submission (page 15).

6. In the methods, in the "Bioorthogonal chemistry reactions" paragraph, explain "MQ"

'MQ' is Milli-Q water. We now explain this first in the "Bioorthogonal chemistry reactions" method sub-section, and only subsequently use the 'MQ' abbreviation (page 16).

7. In the methods, in the "PhosID sample preparation for LC-MS/MS" paragraph

- a. pH is missing for many buffers. For example, 50 mM ABC and 8 M Urea. Also specify in what solution the ABC and Urea Buffer are prepared.
- b. Describe the final conc. of DTT and IAM.
- c. Automated Fe(III)-IMAC workflow, provide essential description here instead of just citing two papers.

Dithiothreitol and iodoacetamide were used at final concentrations of 2mM and 4mM respectively. Urea and ammonium bicarbonate buffer was prepared in Milli-Q water at pH8. These are now clarified in the "PhosID sample preparation for LC-MS/MS" methods sub-section. We now also add

Fe-IMAC conditions to the Methods (page 16).

8. In the methods, in the “LC-MS/MS” paragraph, some key parameters are missing or incomplete. For example, “dynamic exclusion duration of 16 sec” did not specify the times. Collision energy is also missing.

Full MS settings are now added in the same “LC-MS/MS” section of the methods (page 17).

9. Fig3A, fix the typo “Dephosphorlyate”.

We thank the reviewer for pointing this out. We have rectified this in the amended Figure 3A.

10. Fig3D, define the correlation, pearson or spearman? R or R-square? What is LC-MS intensity? Is it peptide or protein intensity? Or just use iBAQ intensity instead of LC-MS intensity since iBAQ intensity is explained in the methods.

Figure 3D features Protein intensities correlated by Pearson linear regression model, and R^2 values are reported. We amended Figure 3D and supplemented the figure legend to make these clearer.

11. In the discussion it said the experiment in Fig4 used one (P-alkyne) of the probes. This should also be clearly indicated in Fig4a instead of using an ambiguous saying “PhosID protocol”.

In the amended Figure 4A workflow schematic, we now specify the approach as “PhosID protocol with P-alkyne”.

12. Fig S5, what’s the “log” in the y-axis label?

We think the reviewer meant Supplementary Figure 6 here. X-axis of Figure S6 was plotted in log10 scale. This is now clarified in the X-axis label as “Unique peptides per protein (Log 10 scale)”.

13. Fig S7, define the correlation. R or R-squared? Pearson or spearman?

Correlation was calculated based on the Pearson linear regression model, and correlation coefficient values (R^2) were indicated in Figure S7 sub-panels. This is now also clarified further in the legend to Figure S7.

14. Fig 4c, x axis label is missing. Texts are overlapping with each other.

We now moved the dash-line labels from the top to the x-axis, as suggested, and text labels are now spaced further apart. Please see the updated Figure 4C.

15. Table S4

a. In the “content” tab, “T.test: t.test based on three technical replicates, two-tailed, paired”. Is it p value or t statistics? Or log-transformed p value?

b. Fix the header of the last tab, they don’t match the content.

Due to the addition of other supplementary tables to address questions above, the contents of Table S4 is now in Supplementary Table 7. We have rectified the header shifts, and given more descriptive details of the column data presented, including the “Student’s T test -log10 p-value”. Please see new version of Supplementary Table 7 that is now submitted together with our response to reviewers.

16. Fig2A, define the black part (labeled by the by-product?)

We are grateful to the reviewer for pointing this detail out. Lines in the original Figure 2A were indeed too thick, such that small blue parts in the stacked bars were difficult to read. The small section in question does not correspond to labeling by the byproducts, but are azide- or alkyne-labeled and should appear as blue. This is now rectified in the new Figure 2.

17. Fig5a and Fig5b, are ratios capped at 5 and -5?

Colour scale is not capped. The colour range is in log₂ scale, from -5 to 5. This is a spread of linear spread of 0.03125 fold (2^{-5}) to 32 fold (2^5). All ratios fall within this range, and are shown without capping.

REVIEWERS' COMMENTS:

Reviewer #1 (Remarks to the Author):

The authors addressed my questions and comments, and the manuscript is suitable for publication. I would have preferred to see an actual experimental comparison with desthiobiotin, but this is outside of the authors' control due to current events. As a suggestion to the editor, other reviewer, and authors about the name issue, Pn is short for phosphonate, so PhosID could easily be switched to PnID.

Reviewer #2 (Remarks to the Author):

The authors have addressed most of the comments. However, there are still a few areas that need to be clarified.

1. About major comment 3/4: It is difficult to carry out new experiments under current situation. But the authors should at least comment on the three phosphonate-containing probes about their potential advantages and disadvantages over each other. For example, it seems proteins are not fully denatured during the labeling process as described in the methods and indicated in Fig1C. Fig1B also shows that three P-handles are quite different in size. Are they subject to steric hindrance to different extent?

The idea of adding a phosphonate handle to existing probes is not new. For example, CPT probes (phosphonate handle + succinimidyl iodoacetamide) were adopted to study free cysteine-containing peptides and protein redox regulation (<https://doi.org/10.1016/j.cell.2020.02.012>). One of the phosphonate handles in this reference is the same as what the authors used in this manuscript. If the authors intend to highlight the three phosphonate-containing probes instead of the existing concept of combining phosphonate handle and other probes, I think these three phosphonate-containing probes should be described in more details to facilitate the adoption by other people in the field.

2. To address major comment 7, the authors added the explanation for "less than 2%" in supplementary figure 1 legend in the revised manuscript. And the newly added explanation says "less than 2.5%". It's a small issue but it's better to be consistent (2% or 2.5%?), and it will be much more clear if the authors specify the 2.5% as what they did for other percentages in line 105 page 5, like "green:blue ratio in input".

3. About the major comment 8, the labeling time is 24 hr vs 3 weeks. Proteins are supposed to undergo turnovers over the longer period. Proteins are not just being synthesized as AHA-containing NSPs, NSPs are also degraded over time. The authors should elaborate more (and be specific) in the manuscript on what the high correlation means, and be specific about the number of proteins that are shared by the two labeling times and unique to either of the labeling time as well. In addition, it seems tab "PhosID - AHA (pulse) - mod" and tab "PhosID - AHA (stable) - mod" are identical in the corresponding table S4.

4. The point of adding a Jurkat result similar to fig4C is to show the readers how the Jurkat data look like overall before comparing two cell lines. For example, do the two cell line data have the same depth? Do they have similar number of significant changes overall?

5. The authors modified Fig1C and it is much better now. The illustration of a 3D protein structure is kept to represent proteins. The description about this part in the methods is a little fragmented. For example, line 492 page 15, cell pellets were resuspended in 1× PBS, pH 7.5, 1% sodiumdeoxycholate (SDC), then diluted to 5 ug/ul in 1× PBS, pH 7.5. Then line 500 page 16, 500 ug total proteins containing 2 M Urea (final) in 1× PBS (pH 7.5) were used for chemistry reactions. Did the authors do a buffer exchange to switch from the lysis buffer to 2 M Urea? Some proteins remain undenatured in 2

M Urea. Is this the reason why the 3D protein structure is kept? Is there a reason why not fully denatured proteins are used in the P-handle reactions? If it's true that proteins are not fully denatured during P-handle reactions, it is better to make it clear in the manuscript.

6. The authors added corrected p values to the IFN experiment. Please state in the methods how the FDR correction was performed.

7. Fig S6, could the authors check on the "Log 10 scale" in the y-axis label? The y-axis is the number of unique peptides per protein, the numbers are quite big if they are on a log10 scale.

8. Fig S7B, the title does not match the text in the figure legend.

9. Fig4C, the authors did add the x-axis text (2-fold, 5-fold, etc). But they don't match the x-axis labels (log2 ratio).

REVIEWERS' COMMENTS:

Reviewer #1 (Remarks to the Author):

The authors addressed my questions and comments, and the manuscript is suitable for publication. I would have preferred to see an actual experimental comparison with desthiobiotin, but this is outside of the authors' control due to current events. As a suggestion to the editor, other reviewer, and authors about the name issue, Pn is short for phosphonate, so PhosID could easily be switched to PnID.

We thank the reviewer for the suggestion, but think PnID is hard to pronounce as an acronym. We still prefer to retain the name 'PhosID', if the Editor is agreeable.

Reviewer #2 (Remarks to the Author):

The authors have addressed most of the comments. However, there are still a few areas that need to be clarified.

1. About major comment 3/4: It is difficult to carry out new experiments under current situation. But the authors should at least comment on the three phosphonate-containing probes about their potential advantages and disadvantages over each other. For example, it seems proteins are not fully denatured during the labeling process as described in the methods and indicated in Fig1C. Fig1B also shows that three P-handles are quite different in size. Are they subject to steric hindrance to different extent?

The idea of adding a phosphonate handle to existing probes is not new. For example, CPT probes (phosphonate handle + succinimidyl iodoacetamide) were adopted to study free cysteine-containing peptides and protein redox regulation (<https://doi.org/10.1016/j.cell.2020.02.012>). One of the phosphonate handles in this reference is the same as what the authors used in this manuscript. If the authors intend to highlight the three phosphonate-containing probes instead of the existing concept of combining phosphonate handle and other probes, I think these three phosphonate-containing probes should be described in more details to facilitate the adoption by other people in the field.

We now expand this with a discussion paragraph on page 9, acknowledging potential differences between the three probes exemplified in this manuscript. Though CPT probes exist, to our knowledge, such probes were never coupled to IMAC purifications for efficient and high throughput processing. Therefore we still think the PhosID approach we describe here offers a significantly novel option with much wider applicability.

2. To address major comment 7, the authors added the explanation for "less than 2%" in supplementary figure 1 legend in the revised manuscript. And the newly added explanation says "less than 2.5%". It's a small issue but it's better to be consistent (2% or 2.5%), and it will be much more clear if the authors specify the 2.5% as what they did for other percentages in line 105 page 5, like "green:blue ratio in input".

We now clarify the 2.5% on page 5 of the main manuscript. The calculations involved to arrive at this 2.5% is also now provided in the Source Data file (tab for Figure 2A and SI Figure 2A).

3. About the major comment 8, the labeling time is 24 hr vs 3 weeks. Proteins are supposed to undergo turnovers over the longer period. Proteins are not just being synthesized as AHA-containing

NSPs, NSPs are also degraded over time. The authors should elaborate more (and be specific) in the manuscript on what the high correlation means, and be specific about the number of proteins that are shared by the two labeling times and unique to either of the labeling time as well. In addition, it seems tab “PhosID - AHA (pulse) – mod” and tab “PhosID - AHA (stable) - mod” are identical in the corresponding table S4.

NSP degradation requires a different experimental setup and reversed stable labeling at the start of the experiment. We therefore cannot speculate about NSP degradation from this experiment that was designed to track only the synthesis of new proteins. In Figure 3D, we compared 24h- and 3h-labeling of (mostly) housekeeping proteins under resting conditions. there is almost complete overlap in the identity of proteins detected at 24h or 3h. This huge overlap is also not surprising, as without any stimulus, the cell is just making the same proteins in routine turnover.

Supplementary Table 4 is now rectified, we thank the reviewer for spotting the duplicated tab. That was not intentional.

4. The point of adding a Jurkat result similar to fig4C is to show the readers how the Jurkat data look like overall before comparing two cell lines. For example, do the two cell line data have the same depth? Do they have similar number of significant changes overall?

Newly synthesized proteins upon interferon gamma stimulation were identified in both cell lines to similar depth. In both cell lines, over 2000 proteins were identified with 24h of stimulation and pulsed AHA labeling. In terms of significant changes at 24h, both cell lines show induction of a small number of responsive proteins (about 80 to 100 proteins). Therefore, in both cell lines, our identification depth was more than sufficient to cover <5% of interferon gamma responsive protein synthesis. The other 95% of identifications were largely coming from constant synthesis of housekeeping proteins, that did not vary with interferon gamma stimulation. A summary tab of these numbers is now also provided in Supplementary Table 7. In addition, we also observed differences between the two cell lines in temporal interferon gamma responses, which we had already summarised in the original Figure 5.

5. The authors modified Fig1C and it is much better now. The illustration of a 3D protein structure is kept to represent proteins. The description about this part in the methods is a little fragmented. For example, line 492 page 15, cell pellets were resuspended in 1× PBS, pH 7.5, 1% sodiumdeoxycholate (SDC)), then diluted to 5 ug/ul in 1× PBS, pH 7.5. Then line 500 page 16, 500 ug total proteins containing 2 M Urea (final) in 1× PBS (pH 7.5) were used for chemistry reactions. Did the authors do a buffer exchange to switch from the lysis buffer to 2 M Urea? Some proteins remain undenatured in 2 M Urea. Is this the reason why the 3D protein structure is kept? Is there a reason why not fully denatured proteins are used in the P-handle reactions? If it's true that proteins are not fully denatured during P-handle reactions, it is better to make it clear in the manuscript.

Urea was supplemented to 2M final concentration, no buffer exchange was performed. The efficiency of bioorthogonal reactions usually decreases with increasing concentration of urea. For this reason, such reactions are mostly performed with 2M urea or less, to balance the need for denaturation and efficiency of biorthogonal reaction. It was not intended to retain protein structures partially.

6. The authors added corrected p values to the IFN experiment. Please state in the methods how the FDR correction was performed.

FDR correction was performed in Perseus with 250 randomisation for $FDR < 0.05$. To aid in interpretation of the corrected p-value, parameters used in FDR correction were already stated in the first description tab in Supplementary Table 7, in the last amended version.

7. Fig S6, could the authors check on the “Log 10 scale” in the y-axis label? The y-axis is the number of unique peptides per protein, the numbers are quite big if they are on a log10 scale.

We thank the reviewer for pointing this out. We have rectified the y-axis modifier. Since the y-axis is labelled in log10 scale, a separate mention of (log 10 scale) was not needed. As such the biggest proteins identified, were with about 100 tryptic peptides.

8. Fig S7B, the title does not match the text in the figure legend.

We thank the reviewer for pointing this out. We have rectified this in the amended version.

9. Fig4C, the authors did add the x-axis text (2-fold, 5-fold, etc). But they don't match the x-axis labels (log2 ratio).

These x-axis texts were added to scale. On the \log_2 ratio scale, 2-fold, 5-fold and 10-fold have the values of $\log_2(2)=1$, $\log_2(5)=2.322$ and $\log_2(10)=3.322$ respectively. The x-axis labels were indeed marked at $x=1$, $x=2.322$ and $x=3.322$.